# Factors associated with HIV testing among young females; further analysis of the 2016 Ethiopian demographic and health survey data

**Yibeltal Alemu Bekele** [ID]*, **Gedefaw Abeje Fekadu**

Department of Reproductive Health and Population Studies, School of Public Health, Bahir Dar University, Bahir Dar, Ethiopia

* yibeltalalemu6@gmail.com

## Abstract

### Background

HIV counseling and testing are key to control and prevent the spread of the virus and improve the lives of people living with HIV. Although the risk of acquiring the virus is high, only 27% of young Ethiopian women age 15 to 24 years old were tested and counseled for HIV. This coverage is low to achieve the 90-90-90 goal. Identifying factors associated with low utilization of HIV testing and counseling services among young females (aged 15 to 24 years) is important to identify the barriers and improve uptake. Therefore, this analysis was done to identify factors associated with low utilization of HIV counseling and testing services among young Ethiopian women.

### Methods

The study used the 2016 Ethiopian demographic and health survey data. The data was downloaded from The DHS program with permission. A total of 2661 young women (aged 15 to 24 years) were included in the final model. Data was weighted to consider disproportionate sampling and non-response. A Complex data management technique was applied to consider the complex sampling technique used in the DHS. Multivariable logistic regression was used to identify factors associated with HIV testing among young women.

### Result

Among sexually active young women, 33.5% (95%CI; 30.1, 37.1) were tested for HIV. Young women who attended primary ((AOR 2.8; (95% CI; 2.0, 3.9)), secondary (AOR 4.7; (95% CI; 3.1, 7.3)) or higher education (AOR; 5.6; 95% CI; 2.6, 12.0), those who had multiple sexual partners (AOR 5.5; 95% (CI; 1.3, 23.3)), young women who ever used alcohol (AOR 1.46; 95% (CI; 1.1, 2.0)) and young women who visited health care facilities (AOR 1.8; (95% CI; 1.4, 2.3)) had higher odds of being tested for HIV. On the other hand, young women from the rural areas had lower odds (AOR 0.5; (95% CI; 0.3, 0.7)) of being tested for HIV.

**Data Availability Statement:** For this analysis, we used the 2016 Ethiopian demographic and health survey data set. The data was accessed from The DHS Program website (https://dhsprogram.com/

data/available-datasets.cfm) for free. We do not have special access privileges to this data. All authors can access the data from this website. To get the data, authors should register and log in. When logged in, they will request to state the project title, co-researchers' name and email and a brief description of the study. After that, the researchers continue to select the country and the data set. Within a few days, he/she will get permission to download the data via email. After the permission, the researcher can login and select the specific data with the format he/she wants.

**Funding:** The authors received no specific funding for this work.

**Competing interests:** The authors have declared that no competing interests exist.

## Conclusion

HIV testing among sexually active young women in Ethiopia was low. Educational status, place of residence, alcohol intake, number of sexual partners and visiting health facility 12 months before the survey were found significant predictors of HIV testing. Therefore, the Ethiopian government should encourage girls to complete secondary education to improve HIV testing and counseling. Young women should be encouraged to visit health facilities to improve HIV testing service uptake.

## Introduction

Globally, 36.5 million people were living with HIV. Every day 5,000 people are infected with the virus. Eastern and Southern Africa remained the most affected regions accounting for 53% of people living with HIV [1]. Among people living with HIV, 30% were young people aged 15 to 24 years [2]. Young women accounted for 64% of the total young people (aged 15 to 24 years) living with HIV. The problem is more severe in Sub-Saharan Africa (SSA) [3].

Young people are exposed to HIV at two points of their lives; at the early age of life (due to mother-to-child transmission) and during their adolescence (as a result of their sexual behaviors and gender disparities) [1]. Young women and girls aged 15 to 24 years are disproportionately affected by HIV because of economic, cultural and social disparities in the society [4–7].

HIV counseling and testing services are essential for HIV prevention, treatment, care, and support. In 2016, the United Nations declared to end AIDS at the end of 2020. The declaration endorsed the 90-90-90 target. Increasing access to and uptake of HIV testing is critical to achieve this target [8]. However, around 30% of people living with HIV were not aware of their HIV status globally in 2016 [9].

HIV testing and counseling services uptake among young women aged 15 to 24 years in SSA remained considerably low. Only 15% of them received HIV testing and counseling in 2013 [10]. Studies in Nigeria, Rwanda, Uganda, and Kenya indicated that only 12%, 40%, 26.5% 27.7% of young women were tested for HIV respectively [11–14].

Studies conducted in different parts of the world identified those young women (aged 20–24 years), those who attended primary or higher-level of education, those who were married, and young women with better socio-economic status (young women with middle, richer and richest wealth index) had higher odds of being tested for HIV. On the other hand, young women living in rural areas had lower odds of being tested for HIV [15–17]. Young women who had multiple sexual partners, those who ever had been pregnant, those who started sexual intercourse after 15 years old, those who had good knowledge about HIV, those who discussed about HIV with mother or female guardian, those who had history of sexually transmitted infections and those who attended antenatal care had higher odds of being tested for HIV [18–23].

Ethiopia is one of the Sub-Saharan countries affected by HIV. The government planned to end the epidemic at the end of 2030 [24, 25]. Besides, the ministry of health planned to end new infections among newborns at the end of 2020 [26]. But the HIV prevalence increased by 10% from 0.30/1000 population in 2008 to 0.33/1000 population in 2016 [27]. Although HIV counseling and testing was one of the strategies designed to achieve the 2020 goal (ending new HIV infections among newborns), only 36.4% of young women were tested for HIV [28].

Identifying factors associated with HIV testing will help to understand the bottlenecks for HIV prevention. Therefore, this analysis was done to identify factors associated with HIV

testing among young women in Ethiopia. The information obtained from this analysis may be used by the ministry of health and other organization working on HIV prevention.

## Methods and materials

### Data

For this analysis, we used the 2016 Ethiopian demographic and health survey (EDHS) data. The 2016 EDHS was a community based, nationally representative data collected from January 18, 2016, to June 27, 2016. The data was collected by the Central Statistics Authority (CSA) and ICF international. The data were downloaded from The DHS program after permission. EDHS followed two stages stratified random sampling technique. A total of 15,683 reproductive age (15 to 49 years old) women were included in the survey. From these 6,401 were aged 15 to 24 years. Among these women, 2,661 had a history of sexual intercourse within 12 months before the survey. Only these women were included in the analysis because they were at risk of contracting HIV and to reduce recall bias.

### Variables

The outcome variable for this study was HIV testing; a dichotomous variable coded as "1" when a young woman reported that she was tested for HIV in the last 12 months before the survey and "0" when she reported otherwise (never tested or tested before 12 months). All reproductive age women included in the survey were asked whether they were tested for HIV or not. The information about the dependent variable was generated from this question.

The independent variables were categorized into two groups; socio-demographic and behavioral. The socio-demographic variables were age, religion, residence, wealth index, educational status, marital status and work status at the time of the survey. Behavior related variables were discussion about HIV with mother or female guardian, history of pregnancy, age at first sexual debut, history of sexually transmitted infections, number of sexual partners, substance use and history of antenatal care.

### Statistical analysis

Data analysis was done using STATA 15.1. After the data set was downloaded from The DHS program, the dependent and independent variables were identified. First, descriptive analysis was done for each variable. Bivariate regression analysis was done to examine associations between HIV testing and the selected predictor variables. Variables that were significant at P-value <0.2 in the bivariate model were included in the multivariable logistic regression model. Besides, multi-collinearity among predictor variables was assessed using the variance inflation factor before recruiting variables to the final mode. Multivariable logistic regression analysis was done to identify factors associated with the outcome variable after adjusting for potential confounders. We weighted the data when computing proportions to consider the non-response and disproportionate sampling used in the DHS sampling process. Since DHS used two stages stratified random sampling technique, complex data analysis techniques were employed when computing standard errors and confidence intervals.

### Ethics approval and consent to participate

The 2016 EDHS protocol was reviewed and approved by the National Ethics Review Committee of the Federal Democratic Republic of Ethiopia, Ministry of Science and Technology and the Institutional Review Board of ICF International. Interviewers collected blood specimens from people who consented for HIV testing. The protocol for blood specimen collection and

analysis was based on an anonymously linked protocol developed for the DHS Program. HIV test results were merged with the socio-demographic data collected in the individual questionnaires after the removal of all information that could potentially identify an individual. The data were anonymous when we accessed it. We received a permission letter from the DHS program to access and use the data.

## Result

A total of 2661 young, sexually active women were included in the final model. One thousand eight hundred ninety-eight (73.3%) of the respondents were aged 20 to 24 years. One thousand two hundred thirteen (45.6%) of the respondents were Muslims in terms of religion, 985 (37.0%) of respondents were Orthodox region followers and 463(17.4%) of respondents were followers of Catholic, protestant and traditional religion. One thousand one hundred eighty-eight (44.6%) of the respondents attended primary education. Two thousand three hundred fifteen (87.0%) of the respondents were married. One thousand nine hundred twelve (75.8%) of the respondents were not working at the time of the survey. One thousand nine hundred thirty (72.5%) of the respondents were rural residents. Two thousand three hundred sixty-eight (89.0%) of the respondents reported that they started sexual intercourse before the age of 20. Three hundred thirteen (11.8%) respondents reported that their most recent sex was non-spousal. One thousand four hundred three (52.7%) respondents visited a health facility within 12 months before the survey. Eight hundred seven (30.3%) of the respondents reported that they had ever drunk alcohol. Twelve (3.5%) respondents reported that they had sex in return for gifts and cash (Table 1).

### HIV testing

Among young women who had a history of sexual intercourse within the last 12 months, only 33.5% (95% CI; 30.1, 37.1) were tested for HIV. HIV testing among young women varied across regions. The proportion of young women tested for HIV from Amhara (9.76%), Oromia (9.38%) and SNNPR (5.74%) regions was high. On the other hand, the proportion of young women tested for HIV from Harari (0.13%), Gambela (0.2%) and Dire Dewa (0.3%) regions were low.

### HIV testing by women characteristics

Among those tested for HIV, 26.2% were aged 15 to 19 years, 15.1% did not attend formal education, 81.7% were married or living in union, 46.2% were urban residents and 66.0% visited health facility within 12 months before the survey (Table 2).

### Factors associated with HIV testing

Age at the time of the survey, educational status, age at first sex, marital status, residence, working status at the time of the survey, the number of sexual partners, history of alcohol uptake and health care facilities visit in the last 12 months before the survey were recruited to the multivariable logistic regression model. In the multivariable logistic regression model, the number of sexual partners, educational status, marital status, and residence, history of health facility visit 12 months before the survey and history of drinking alcohol were found significantly associated with HIV testing.

The odds of being tested for HIV among young women who attended primary and secondary education was 2.78 (AOR 2.78; (95% CI; 2.01, 3.87)) and 4.73 (AOR 4.73; (95% CI; 3.07, 7.29)) respectively higher compared to those who did not attend formal education. The odds

**Table 1. Socio-demographic characteristic of young, sexually active women in Ethiopia, EDHS 2016 (N = 2661).**

| Characteristics | Number (%) |
|---|---|
| **Age at the time of the survey** | |
| 15 to 19 | 763 (28.7) |
| 20 to 24 | 1898(73.3) |
| **Educational status** | |
| No formal education | 894(33.6) |
| Primary | 1188(44.6) |
| Secondary | 405(15.2) |
| Higher | 174(6.5) |
| **Marital status** | |
| Single | 187(7.0) |
| Married or living in union | 2315(87.0) |
| Other** | 159(6.0) |
| **Age at first birth** | |
| Before 20 years | 1315(76.3) |
| At 20 and after years | 410(23.8) |
| **Wealth index** | |
| Poorest | 813(30.6) |
| Poor | 383(14.4) |
| Middle | 345(13.0) |
| Richer | 312(11.7) |
| Richest | 808(30.3) |
| **Residence** | |
| Urban | 731(27.5) |
| Rural | 1930(72.5) |
| **Age at first sex** | |
| Before 20 years | 2368(89.0) |
| At 20 and after years | 293(11.0) |
| **Numbers of lifetime sexual partners** | |
| One | 2636(99.0) |
| More than one | 25(1.0) |
| **Visited health facility in the last 12 months before the survey** | |
| Yes | 1403(52.72) |
| No | 1258(47.28) |
| **Chewed khat** | |
| No | 2426(92.0) |
| Yes | 235(8.83) |
| **Ever drunk alcohol** | |
| No | 1854(69.7) |
| Yes | 807(30.3) |

Other** = divorced and widowed

of being tested for HIV among young women who were widowed, divorced and separated was 2.18 times higher compared to young single women (AOR 2.18; (95% CI; 1.01, 4.71)). Young women who were living in a rural areas had 63.0% (AOR 0.47; (95% CI; 0.31, 0.72)) lower odds of being tested for HIV compared to their urban counterparts.

The odds of being tested for HIV among sexually active, young women who had multiple sexual partners was 5.49 (AOR 5.49; 95% (CI; 1.29, 23.27)) times higher compared to young

**Table 2. Characteristics of HIV tested, sexually active, young women in Ethiopia, EDHS 2016 (N = 1003).**

| Characteristics | Number (%) |
|---|---|
| **Age at the time of the survey** | |
| 15 to 19 | 263(26.2) |
| 20 to 24 | 740(77.8) |
| **Religion** | |
| Orthodox | 505(50.4) |
| Muslim | 327(32.6) |
| Other* | 171(17.0) |
| **Educational status** | |
| No formal education | 151(15.1) |
| Primary | 481(48.0) |
| Secondary | 246(24.5) |
| Higher | 125(12.5) |
| **Marital status** | |
| Single | 116(11.6) |
| Married or living in union | 820(81.8) |
| Other** | 67(6.6) |
| **Age at first birth** | |
| Before 20 years | 396(69.0) |
| At 20 and after years | 178(31.0) |
| **Working status at the time of the survey** | |
| No | 648(64.6) |
| Yes | 355(35.4) |
| **Wealth index** | |
| Poorest | 125(12.5) |
| Poor | 121(12.1) |
| Middle | 114(11.4) |
| Richer | 136(13.6) |
| Richest | 507(50.6) |
| **Residence** | |
| Urban | 463(46.2) |
| Rural | 540(53.8) |
| **Age at first sex** | |
| Before 20 years | 841(83.9) |
| At 20 and after years | 162(16.1) |
| **Visited health facility within 12 months before the survey** | |
| Yes | 662(66.0) |
| No | 341(34.0) |

Other* = Protestant, Catholic and traditional other** = divorced and widowed

women who had one sexual partner. The odds of being tested for HIV among young women who ever had drunk alcohol was 1.46 (AOR 1.46; 95% (CI; 1.09, 1.98)) times higher compared to those who had not. The odds of being tested for HIV among young women who visited a health facility in 12 months before the survey was 1.78 (AOR 1.78; (95% CI; 1.36, 2.32)) times higher compared to those young women who never drunk (Table 3).

**Table 3. Factors associated with HIV testing among sexually active, young women in Ethiopia, EDHS 2016 (N = 2661).**

| Variables | HIV test | | COR (95% CI) | AOR (95% CI) |
|---|---|---|---|---|
| | Yes | No | | |
| **Age at the time of the survey** | | | | |
| 15 to 19 | 263 | 500 | 1 | 1 |
| 20 to 24 | 740 | 1158 | 1.21(0.91, 1.61) | 1.14(0.84, 1.56) |
| **Educational status** | | | | |
| No formal education | 151 | 743 | 1 | 1 |
| Primary education | 481 | 707 | 2.95(2.13, 4.09) | 2.78(1.01, 3.86) |
| Secondary education | 246 | 159 | 6.88(4.39, 10.77) | 4.73(3.07, 7.29) |
| Higher | 125 | 49 | 10.05(5.09, 19.84) | 5.55(2.56, 12.02) |
| **Age at first sex** | | | | |
| Before 20 years | 841 | 131 | 1 | 1 |
| At 20 and after | 162 | 162 | 1.87(1.25, 2.80) | 1.11(0.96, 1.78) |
| **Marital status** | | | | |
| Single | 116 | 71 | 1 | 1 |
| Married/ in union | 820 | 1495 | 0.45(0.27, 0.77) | 1.31(0.74, 2.35) |
| Other | 67 | 92 | 0.73(0.34, 1.62) | 2.18(1.01, 4.71) |
| **Residence** | | | | |
| Urban | 463 | 268 | 1 | 1 |
| Rural | 540 | 1390 | 0.25(0.16, 0.38) | 0.47(0.31, 0.72) |
| **Working status at the time of survey** | | | | |
| No | 648 | 1264 | 1 | 1 |
| Yes | 355 | 394 | 1.72(1.26, 2.35) | 1.12(0.80, 1.57) |
| **Numbers of sexual partners** | | | | |
| One | 989 | 1647 | 1 | 1 |
| More than one | 14 | 11 | 7.44(2.23, 24.84) | 5.49(1.29, 23.27) |
| **Ever taken alcohol** | | | | |
| No | 617 | 1237 | 1 | 1 |
| Yes | 386 | 421 | 1.66(1.24, 2.23) | 1.46(1.09, 1.98) |
| **Visiting health facility in the last 12 months** | | | | |
| No | 341 | 917 | 1 | 1 |
| Yes | 662 | 741 | 2.03(1.57, 2.62) | 1.77(1.36, 2.32) |

Other = divorced and widowed

## Discussion

The magnitude of HIV testing among sexually active young women (who had sexual intercourse 12 months before the survey) was 33.5% (95% CI; 30.1, 37.1). This finding was consistent with a study conducted in South Africa which showed that 32.7% of young females (aged 15–24 years) were tested for HIV [29]. However, the proportion of young women tested for HIV in this study was lower than studies conducted in Uganda (92%) [30], Karamoja region (81.8%) [31] and another study conducted in South Africa (60.1%) [16]. The reason for this difference might be the difference in the modalities the Uganda government used to reach the community for HIV counseling and testing (HCT). The Uganda government integrated HCT to routine health care service at all levels, expanded community outreach or mobile HCT services targeting the grass root level and hard to reach areas and gave strong emphasis on provider-initiated HCT to minimize missed opportunities [32]. Uganda is one of the few

Sub-Saharan countries which effectively implemented a home-based HCT service which played a major role in expanding access to HCT [33]. The Ethiopian government may take this lesson to improve HIV testing service uptake in the country.

This study identified that young women who attended formal education had more odds of being tested for HIV. This finding was consistent with studies conducted in Tanzania [15, 34] and Nigeria [22, 23]. The reason for this is that education can improve HIV knowledge. Education also empowers women to make decisions to visit the health facility and use health services. Besides, education improves income among women which in turn increases health service use [35]. Strengthening the available initiatives to enable all Ethiopian girls to attend primary, secondary or higher level of education may help to improve HIV testing service uptake.

Young women who were living in rural areas had 53.0% lower odds of being tested for HIV compared to their urban counterparts. A similar conclusion was drawn from studies conducted in Ethiopia [36] and Nigeria [22, 23]. The reason for this may be better availability and accessibility of HIV testing facilities in urban settings [37]. Ethiopia should make HIV testing facilities and services more accessible to the rural community.

Young women with multiple sexual partners were more likely to be tested for HIV compared to those with one sexual partner. This finding was similar to studies conducted in Tanzania [15], Uganda [30] and Thailand [19]. This might be due to the fact that women with multiple sexual partners had a higher perceived risk of acquiring HIV. This, in turn, increases their motive to be tested [38]. Moreover, program planners and health professionals give emphasis to these women considering them high risk.

Young women who ever drunk alcohol had higher odds of being tested for HIV compared to women who never drank. This could be due to risky sexual behavior after alcohol. This may have increased perceived susceptibility to HIV which in turn leads them to be tested for HIV [39, 40].

This study showed that women who visited health facilities 12 months before the survey had more odds of being tested for HIV compared to young women who did not visit health facilities. This finding was similar to a study conducted in South Africa [29]. This might be due to the fact that health professionals initiate people who visited health facilities for HIV testing. The service is provided at all governmental and public health facilities in Ethiopia [41]. Ethiopia adopted provider-initiated counseling and testing (PICT) for all outpatient and inpatient clients [41, 42]. Encouraging young women to visit health facilities may increase HIV testing uptake.

The strength of this analysis is that it was based on nationally representative data with a large sample size. However, recall and social desirability biases may have affected the results. To reduce recall bias, we restricted the analysis to young women age 15 to 24 that had a history of sexual intercourse within the last 12 months before the survey.

## Conclusion

HIV testing among sexually active young women in Ethiopia was low. Educational status, place of residence, history of alcohol intake, number of sexual partners and visiting health facility 12 months before the survey were found significant predictors of HIV testing among sexually active young women in Ethiopia. The Ethiopian government needs to intensify efforts to expand education for all girls. Improving access to HIV testing for rural women may also increase HIV testing services uptake. Besides, encouraging young women to visit Health facilities is important to increase the proportion of women tested for HIV and achieve the 90-90-90 target.

## Acknowledgments

The authors would like to thank the DHS program for providing the data.

## Author Contributions

**Conceptualization:** Yibeltal Alemu Bekele, Gedefaw Abeje Fekadu.

**Data curation:** Yibeltal Alemu Bekele, Gedefaw Abeje Fekadu.

**Formal analysis:** Yibeltal Alemu Bekele.

**Investigation:** Yibeltal Alemu Bekele, Gedefaw Abeje Fekadu.

**Methodology:** Yibeltal Alemu Bekele, Gedefaw Abeje Fekadu.

**Software:** Yibeltal Alemu Bekele, Gedefaw Abeje Fekadu.

**Writing – original draft:** Yibeltal Alemu Bekele, Gedefaw Abeje Fekadu.

**Writing – review & editing:** Yibeltal Alemu Bekele, Gedefaw Abeje Fekadu.

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
