## [Decision Letter · Decision Letter 0]

12 Dec 2019

PONE-D-19-32104

Factors associated with HIV testing among young females; further analysis of the 2016 Ethiopian Demographic and Health Survey.

PLOS ONE

Dear Mr Bekele,

Thank you for submitting your manuscript to PLOS ONE. After careful consideration, we feel that it has merit but does not fully meet PLOS ONE’s publication criteria as it currently stands. Therefore, we invite you to submit a revised version of the manuscript that addresses the points raised during the review process.

We would appreciate receiving your revised manuscript by 26th January 2020. To enhance the reproducibility of your results, we recommend that if applicable you deposit your laboratory protocols in protocols.io, where a protocol can be assigned its own identifier (DOI) such that it can be cited independently in the future. For instructions see: http://journals.plos.org/plosone/s/submission-guidelines#loc-laboratory-protocols

We look forward to receiving your revised manuscript.

Kind regards,

Kwasi Torpey, MD PhD MPH

Academic Editor

PLOS ONE

Journal Requirements:

3. In ethics statement in the manuscript and in the online submission form, please provide additional information about the patient records/samples used in your retrospective study. Specifically, please ensure that you have discussed whether all data/samples were fully anonymized before you accessed them and/or whether the IRB or ethics committee waived the requirement for informed consent. If patients provided informed written consent to have data/samples from their medical records used in research, please include this information.

4. Your ethics statement must appear in the Methods section of your manuscript. If your ethics statement is written in any section besides the Methods, please move it to the Methods section and delete it from any other section. Please also ensure that your ethics statement is included in your manuscript, as the ethics section of your online submission will not be published alongside your manuscript.

Additional Editor Comments (if provided):

The authors should review the references and ensure it is consistent with the journal requirements

1. Typo in Reference 7, 10 Correct spelling of "Organazation"

2. Ref 8,9,11,14 - Please change from CAPS

3. Ref 16 - Not consistent with referencing guidelines. Has initial then surname. Please correct

4. Ref 17,18,19, 20 is non compliant with referencing guidelines. Please correct

5. Ref 39: Who is the author?

Reviewers' comments:

Reviewer's Responses to Questions

**Comments to the Author**

1. Is the manuscript technically sound, and do the data support the conclusions?

Reviewer #1: Yes

Reviewer #2: Yes

2. Has the statistical analysis been performed appropriately and rigorously? 

Reviewer #1: No

Reviewer #2: Yes

3. Have the authors made all data underlying the findings in their manuscript fully available?

Reviewer #1: Yes

Reviewer #2: No

4. Is the manuscript presented in an intelligible fashion and written in standard English?

Reviewer #1: Yes

Reviewer #2: No

5. Review Comments to the Author

Reviewer #1: The authors need to address the comments raised in the summary of the review and in the manuscript as per attachments before the manuscript being considered for publication

Section Comment, question, suggestion.

Abstract 1. Well summarised abstract.

2. Lacked quantifications of the problem, how low is low HIV testing in Ethiopia?

3. Regarding the knowledge gap:

o What are the advantages of us having this information? Who will benefit from the study being conducted and how?

o Limited information does not warrant a study to be conducted. The missing information has to be useful in some way to the study stakeholders.

4. Would be better to mention the age range. Who is considered to be a young woman for this study? 15-19 years? Or 15-24?

5. The conclusion summarized the results without any further recommendations.

o Recommendations are needed to show the implication of the study findings either to policy and/or to practice.

Introduction

Background 1. Throughout the entire background, entire paragraphs had only one citation on the last sentence. Please revisit citations.

2. It is still unclear who is a “young woman”. It is informative for the reader to have an age range earlier on in the paper.

3. For the paragraph concerning previously identified significant predictors;

o It would be more informative to mention the exact levels/categories which were found to be significant predictors e.g. Rural/urban? Which category was significant?

o It would also be informative to mention whether they are predictors of higher/lower odds of HIV testing.

4. And thereafter identifying the factors, how would the results be used? After these bottle-necks have been identified.

Methodology 1. Why only include women who were sexually active in the past 12 months into the study? Please provide an elaboration.

o The risk of HIV does not change regardless of the time since last sexual intercourse.

o Wouldn’t a more appropriate exclusion criteria be women who NEVER had sexual intercourse?

2. For the outcome definition;

o Why the focus on those who tested in past 12 months?

o Why not use if she EVER tested for HIV?

o Why are those who tested before 12 months considered as if they have never tested and coded as 0?

3. How was HIV knowledge measured?

4. Did you consider using alternatives to logistic regression if you identified the outcome of interest to be common? >10% prevalent.

5. “Bivariate regression analysis was done to examine associations between contraceptive use and the selected predictor variables”;

o Does this study measure contraceptive use OR HIV testing?

6. “In addition, correlation among predictor variables was assed using variance inflation factor before recruiting variables to the final mode.”

o What do you mean? Is it correlation or collinearity?

Results 1. For every table caption, please include N=???

2. For every variable in the tables with category “other”, please provide an explanation on who/what “other” means.

3. Did you consider life time sexual partners? Or those in the past 12 months? Please make clear.

4. What is chat? Concerning the variable “chewed chat”

5. The percentages presented in Table 3 are column percentages,

o Which means the interpretations should be “among those tested for HIV, 26.22% were aged 15-19”.

o Please revise the interpretation provided above.

Discussion Please apply these for the entire discussion section;

1. For each finding, it would be more informative to report numbers/findings that these previous studies reported. It would be informative to mention them in brief so the reader may compare.

2. What is the implication of each study finding? Are you recommending for the Ethiopian government to adopt a similar strategy to increase HIV testing as Uganda?

3. Discussing only similarities/differences of study findings is not enough. You need to go a step further and show the implications of this study findings to either policy or to practice.

Strengths and limitations 1. This section is missing from the paper

Conclusion + Recommendations 1. The conclusion provided a summary of the main results of the paper.

2. Please provide recommendations basing on the results of this study. How can these findings be applied to benefit the stakeholders of the study? The women, policy makers, government officials?

Reviewer #2: Manuscript Number: PONE-D-19-32104

Full Title: Factors associated with HIV testing among young females; further analysis of the 2016

Ethiopian Demographic and Health Survey.

Review Comments

The manuscript presents the very interesting and useful study that is very crucial to inform the improvement in HIV counselling and testing in Ethiopia and other sub-Saharan countries. The authors presented straight forward findings that are easy to follow. However, authors need to revise the manuscript per the below recommendations. The writing style, mainly on structuring the paragraphs, gramma and typos may need to be given more attention in the whole manuscript.

Abstract.

1. Well structured,

2. Few grammatical errors exist and one statements (e.g ….. those who ever had alcohol (AOR 1.46; 95% (CI; 1.09, 1.98)) and young women who visited health facility (AOR 1.78; (95% CI; 1.36, 2.32)) higher odds of being tested for HIV.) has a word missing

Background

3. Paragraph 3, last statement needs further elaboration to put context. It is unclear as to which population is represented by the 30 % of people living with HIV.

4. Citations need to be specific to the statements rather than lumping all the references to the last statement of the paragraph.

5. The last paragraph of introduction indicates that there are few studies, on factors associated with HIV testing among young women in Ethiopia. Unless the gap is identified in those studies, the current study is unjustifiable. The authors may need to present what those studies found and their gaps to justify why the current study is needed.

6. The authors may need to revise the whole introduction section and correct the grammatical, spelling, and space errors.

Methods and Materials

7. Statistical analysis needs to be described clearer. The authors indicate that “Bivariate regression analysis was done to examine associations between contraceptive use and the selected predictor variables. Contraceptive use was not listed as an outcome variable in the above section. Authors may need to check if this is correct.

8. The procedure for weighting the data to account for non-response and disproportionate sampling need to be transparently described.

Results

9. The results are well written; however, the authors may consider making the results more concise. It is possible to fuse tables 1 and 3; and 2 and 3 by considering that the numbers in table 3 are subsets of Tables 1 and 2.

10. The authors may also need to indicate which variables were adjusted in the multivariate model, what criteria were used for selecting such variables for adjusting and/or justification.

11. The data for crude odds ratio for rural residency in table 4 is missing

12. The authors may also want to reformat the Table 4 so that numbers separated by comma are spaced.

Discussion

13. The discussion has interpreted and compared the findings with the previous studies. However, the authors may need to restructure the discussion a bit, so they begin the section by summarizing what they found and later discussing the results.

14. Discussion of the methodological strengths and weaknesses/limitations of their study is missing

15. Although they found the factors associated with HIV testing among sexually active young women in Ethiopia, their discussion needs to translate and discuss the findings by relating to the real issues among young women in Ethiopia.

16. To improve the success of the HIV counselling and testing program, the authors need to indicate the implications for practice and further research.

6. PLOS authors have the option to publish the peer review history of their article (what does this mean?). If published, this will include your full peer review and any attached files.

Reviewer #1: Yes: Michael Johnson Mahande

Reviewer #2: Yes: MASIKA, Golden Mwakibo

---

## [Author Response · Author response to Decision Letter 0]

10 Jan 2020

Point by point response

S/N Reviewers’ /editor’s comments Authors response 

 Editor’s comments 

1. Typo in Reference 7, 10 correct spelling of "Organazation" Thank you for the comment. We correct the spellings of “organization” on the updated manuscript. (line number 35-and page numbers)

2. Ref 8, 9, 11, 14 - Please change from CAPS Thank you for the comment. We change CAPS on the updated manuscript. (line number 259 and page number 13)

3. Ref 16 - Not consistent with referencing guidelines. Has initial then surname. Please correct Thank you for the suggestion. We accept and amend the correction on the updated manuscript 

(line number 310 and page number 13)

4. Ref 17, 18, 19, 20 is non-compliant with referencing guidelines. Please correct Thank you for the comment. We made correction and in line with the referencing guide line. (line numbers 314-322 and page number 14)

5. Ref 39: Who is the author? Thank you for the comment. We correct it “Hopkins J. Drinking and Risky Sexual Behavior 2015”

(line number 364 and page number 15)

Reviewer 1

Abstract section 

1. Lacked quantifications of the problem, how low is low HIV testing in Ethiopia? Thank you for the comment. We quantified the problem specially in Ethiopia context 

(line number 35 and page number 2 )

2. Regarding the knowledge gap:

a. What are the advantages of us having this information? Who will benefit from the study being conducted and how?

b. Limited information does not warrant a study to be conducted. The missing information has to be useful in some way to the study stakeholders. Thank you for the comment. We revised the introduction to show the importance of conducting the study. In the revision, we included beneficiaries of the study output. (line numbers 36-39 and page number 2)

3. Would be better to mention the age range. Who is considered to be a young woman for this study? 15-19 years? Or 15-24? Thank you for the suggestion. In this study young means those women’s whose age 15-24 years old. And this is included in the revised version. (line numbers 35, 37, 41-42 and page number 2) 

4. The conclusion summarized the results without any further recommendations.

a. Recommendations are needed to show the implication of the study findings either to policy and/or to practice. Thank you for the suggestion. Based on your comment we incorporate recommendation on conclusion section.

(Line numbers 56-58 and page number 2) 

Introduction section 

1. Throughout the entire background, entire paragraphs had only one citation on the last sentence. Please revisit citations. Thank you for the comment. We revise and made correction on the updated manuscript 

(line numbers 65- 107 and page numbers 3and 4)

2. It is still unclear who is a “young woman”. It is informative for the reader to have an age range earlier on in the paper. Thank you for the suggestion. A young woman means, a woman’s whose age between 15 -24 years old.

(line numbers 65-67,72,81 and page number 3)

3. For the paragraph concerning previously identified significant predictors; 

o It would be more informative to mention the exact levels/categories which were found to be significant predictors e.g. Rural/urban? Which category was significant? 

o It would also be informative to mention whether they are predictors of higher/lower odds of HIV testing. Thank you for the comment. Based on the comment we clearly mentioned the level and categories significantly associated with HIV testing. 

(line numbers 86-94 and page numbers 3 and 4)

4. And thereafter identifying the factors, how would the results be used? After these bottle-necks have been identified. Thank you for the comment. The ministry of health and other organizations working on HIV may use to strengthen HIV prevention activities. This idea is included in the revised section of the manuscript. 

(line numbers 103-107 and page number 4)

Methodology section 

1. Why only include women who were sexually active in the past 12 months into the study? Please provide an elaboration.

o The risk of HIV does not change regardless of the time since last sexual intercourse.

o Wouldn’t a more appropriate exclusion criteria be women who NEVER had sexual intercourse?

Thank you for the comment. We included women who had history of sexual intercourse with in the last 12 months. This to reduce recall bias. 

2. For the outcome definition;

o Why the focus on those who tested in past 12 months?

o Why not use if she EVER tested for HIV?

o Why are those who tested before 12 months considered as if they have never tested and coded as 0? 

 Thank you for the comment. We limited the analysis HIV testing for the last 12 months to see the most recent trend. We feel that the most recent information is more important than the earlier ones for HIV prevention. If sexually active, the testing information in the last year is more informative than history of testing before four years. In addition, most of the variables, for example, characteristics of women are at the time of survey. Since the data collection was cross-sectional, events that happened years before the time of survey may have different implications. For example, if we take age, and include young women who were tested 5 years before, the information about age will be quite different. 

3. How was HIV knowledge measured?

 Thank you for the suggestion. This variable was not included in the analysis. Now it is deleted. (line number 128 and page number 5)

4. Did you consider using alternatives to logistic regression if you identified the outcome of interest to be common? >10% prevalent.

 Thank you for the comment. We did not consider because the outcome variable is binary. We feel that this is the best option. 

5. “Bivariate regression analysis was done to examine associations between contraceptive use and the selected predictor variables”;

o Does this study measure contraceptive use OR HIV testing?

 Thank you for the comment and we would like to say sorry for the silly mistake we made. It was to say “HIV testing”. Finally we made amendment on the updated manuscript (line number 135 and page number 5)

6. “In addition, correlation among predictor variables was assed using variance inflation factor before recruiting variables to the final mode.”

o What do you mean? Is it correlation or collinearity? Thank you the comment and We would like to say sorry again for made this silly mistake using “correlation” instead of “collinearity”. Finally we made amendment on the updated manuscript (line number 137 and page number 5)

Result section 

1. For every table caption, please include N=??? Thank you for the suggestion. Based on the comment we included N in the entire table in the updated manuscript.

2. For every variable in the tables with category “other”, please provide an explanation on who/what “other” means. Thank you for the suggestion. Based on the recommendation we provided footnote explanation what “other” means.

3. Did you consider life time sexual partners? Or those in the past 12 months? Please make clear. Thank you for the comment. In this analysis we considered “life time sexual partners” not “those in the past 12 months”. 

4. What is chat? Concerning the variable “chewed chat” Thank you for the constructive comment. Chat is a flowering plant native to the horn of Africa and Arabian Peninsula commonly called as “khat” but it’s commonly called as “chat” in our country Ethiopia. But to make it familiar we change “chat” in to “khat” in the updated manuscript. 

5. The percentages presented in Table 3 are column percentages, 

o Which means the interpretations should be “among those tested for HIV, 26.22% were aged 15-19”. 

o Please revise the interpretation provided above. Thank you for the constructive comments. The interpretation is corrected now. The data for those aged 15 – 19 is presented in the table.

Discussion 

Please apply these for the entire discussion section;

1. For each finding, it would be more informative to report numbers/findings that these previous studies reported. It would be informative to mention them in brief so the reader may compare. Thank you for the constructive comment. Based on your recommendation we incorporated the findings of each study on the updated manuscript.

 (line numbers 204-208 and page number 10)

2. What is the implication of each study finding? Are you recommending for the Ethiopian government to adopt a similar strategy to increase HIV testing as Uganda? Thank you for the constructive comment. Based on the recommendation we incorporated the implications of each finding on the updated manuscript.

(line numbers 213 -2014 and page number 10)

3. Discussing only similarities/differences of study findings is not enough. You need to go a step further and show the implications of this study findings to either policy or to practice. Thank you for the constructive comment. Based on the recommendation we included the implications of each finding on the updated manuscript.

(line numbers 220 – 222, 226-227, 242-243 and page number 10)

Strength and limitation 

1. This section is missing from the paper Thank you for the constructive comment. We included the limitation and strength in the updated manuscript.

(line numbers 244-247 and page number 11)

Conclusion and Recommendation section 

1. Please provide recommendations basing on the results of this study. How can these findings be applied to benefit the stakeholders of the study? The women, policy makers, government officials? Thanks for the constructive comment. Based on your recommendation we included recommendation in the updated manuscript.

(line numbers 252-255 and page number 12)

Reviewer 2 

 Abstract 

1. Few grammatical errors exist and one statements (e.g ….. those who ever had alcohol (AOR 1.46; 95% (CI; 1.09, 1.98)) and young women who visited health facility (AOR 1.78; (95% CI; 1.36, 2.32)) higher odds of being tested for HIV.) has a word missing Thank you for the comment. All grammatical errors and missing statements are corrected in the revised manuscript. 

(line numbers 49-51 and page number)

 Background 

2. . Paragraph 3, last statement needs further elaboration to put context. It is unclear as to which population is represented by the 30 % of people living with HIV. Thank you for the comment. It is revised and made clear now. 

(line numbers 78-79 and page number 3)

3. Citations need to be specific to the statements rather than lumping all the references to the last statement of the paragraph. Thank you for the constructive comments. The citations are revised the revised manuscript. 

4. The last paragraph of introduction indicates that there are few studies, on factors associated with HIV testing among young women in Ethiopia. Unless the gap is identified in those studies, the current study is unjustifiable. The authors may need to present what those studies found and their gaps to justify why the current study is needed. Thank you for the suggestion, the paragraph is revised to show the existing gaps and the importance of the study

(line numbers 103- 107and page number 4)

5. The authors may need to revise the whole introduction section and correct the grammatical, spelling, and space errors.

Methods and Materials Thank you for the comment. We corrected the grammatical, spelling and spacing errors in the revised manuscript. 

 Methods 

6. Statistical analysis needs to be described clearer. The authors indicate that “Bivariate regression analysis was done to examine associations between contraceptive use and the selected predictor variables. Contraceptive use was not listed as an outcome variable in the above section. Authors may need to check if this is correct. Thank you for the comment. We understood that we made editing errors. The dependent variable was HIV testing, not contraceptive use. Now, we revised it. (line numbers 134- 137 and page number 5)

7. The procedure for weighting the data to account for non-response and disproportionate sampling need to be transparently described. Thank you for the comment. EDHS takes stratified random sampling technique. Samples taken from some regions may be high and in other regions low. Therefore, DHS recommend data weighting to reduce the effect of over sampling or under sampling. The details can be find in the DHS website. 

Results 

8. The results are well written; however, the authors may consider making the results more concise. It is possible to fuse tables 1 and 3; and 2 and 3 by considering that the numbers in table 3 are subsets of Tables 1 and 2. Thank you for the comment. We merged table 1 and 2. 

9. The authors may also need to indicate which variables were adjusted in the multivariate model, what criteria were used for selecting such variables for adjusting and/or justification. Thank you for the comment. All variables were which were not multicollinear were included to the multivariate logistic regression model

10. The data for crude odds ratio for rural residency in table 4 is missing Thank you for the comment. It is included in the revised version.

11. The authors may also want to reformat the Table 4 so that numbers separated by comma are spaced. Thank you for the comment. It is formatted now. 

Discussion 

12. The discussion has interpreted and compared the findings with the previous studies. However, the authors may need to restructure the discussion a bit, so they begin the section by summarizing what they found and later discussing the results. Thank you for the constructive comment. The discussion is revised now. 

13. Discussion of the methodological strengths and weaknesses/limitations of their study is missing Thank you for the comment. It is included at the end of the discussion section in the revised manuscript. (line numbers 244-247 and page number 11 )

14. Although they found the factors associated with HIV testing among sexually active young women in Ethiopia, their discussion needs to translate and discuss the findings by relating to the real issues among young women in Ethiopia. Thank you for the comment. 

We tried to translate the findings to the Ethiopian context 

15. To improve the success of the HIV counseling and testing program, the authors need to indicate the implications for practice and further research. Thank you for the comment. We included recommendation in the revised section of the manuscript.

---

## [Editor Report · Decision Letter 1]

15 Jan 2020

PONE-D-19-32104R1

Factors associated with HIV testing among young females; further analysis of the 2016 Ethiopian Demographic and Health Survey.

PLOS ONE

Dear Mr Yibeltal Bekele,

Thank you for submitting your manuscript to PLOS ONE. After careful consideration, we feel that it has merit but does not fully meet PLOS ONE’s publication criteria as it currently stands. Therefore, we invite you to submit a revised version of the manuscript that addresses the points raised during the review process. The manuscript has several language errors which need attention.

We would appreciate receiving your revised manuscript by 30th January 2020. To enhance the reproducibility of your results, we recommend that if applicable you deposit your laboratory protocols in protocols.io, where a protocol can be assigned its own identifier (DOI) such that it can be cited independently in the future. For instructions see: http://journals.plos.org/plosone/s/submission-guidelines#loc-laboratory-protocols

We look forward to receiving your revised manuscript.

Kind regards,

Kwasi Torpey, MD PhD MPH

Academic Editor

PLOS ONE

Additional Editor Comments (if provided):

The manuscript requires significant copyediting before acceptance. The language in its current form is not acceptable. I strongly suggest a fluent native speaker copyedits the document. There are several language errors through the whole document too numerous to recount. I am highlighting a few

Title page: Correspondent should read corresponding

Abstract Line 5. low HIV testing and counseling service use better written low utilization of HIV testing and counseling service

Introduction: Line 1 5,000 people were infected should be 5,000 people are infected

Line 3 disproportionally should be disproportionately

Intro 2nd para Line 1 and 2 "early life of transmitted from mother to child …… needs to rephrase. There are many more in the narrative

---

## [Author Response · Author response to Decision Letter 1]

22 Jan 2020

Point by point response

1. The manuscript requires significant copyediting before acceptance.

Thank you for your suggestion. For addressing the issue we consulted senior public health staff and language professors in my university. We also used online softwares, specifically grammerly and scribens (check for correctness of spellings).

2. Title page: Correspondent should read corresponding

Thank you for your comment. Based on your comment we made amendment 

3. Abstract Line 5. low HIV testing and counseling service use better written low utilization of HIV testing and counseling service

Thank you for your suggestion. Based on your suggestion we rewrite it. 

4. Introduction: Line 1 5,000 people were infected should be 5,000 people are infected

Thank you for your suggestion. Based on your suggestion we rewrite it. 

5. Line 3 disproportionally should be disproportionately

Thank you for your comment. We correct the spelling error on the updated manuscript. 

6. Intro 2nd para Line 1 and 2 "early life of transmitted from mother to child …… needs to rephrase. 

Thank you for your suggestion. Based on your suggestion we rewrite it. 

7. There are many more in the narrative

Thank you for your comment. Based on your comment we revise the rest parts of the document.

---

## [Editor Report · Decision Letter 2]

24 Jan 2020

Factors associated with HIV testing among young females; further analysis of the 2016 Ethiopian demographic and health survey data

PONE-D-19-32104R2

Dear Mr Yibeltal Bekele,

We are pleased to inform you that your manuscript has been judged scientifically suitable for publication and will be formally accepted for publication once it complies with all outstanding technical requirements.

With kind regards,

Kwasi Torpey, MD PhD MPH

Academic Editor

PLOS ONE
---

## [Editor Report · Acceptance letter]

29 Jan 2020

PONE-D-19-32104R2 

Factors associated with HIV testing among young females; further analysis of the 2016 Ethiopian demographic and health survey data 

Dear Dr. Bekele:

I am pleased to inform you that your manuscript has been deemed suitable for publication in PLOS ONE. Congratulations! Your manuscript is now with our production department. 

With kind regards,

on behalf of

Professor Kwasi Torpey 

Academic Editor

PLOS ONE